# A Hybrid Approach for Quantitative Analysis of Fire Hazards in Enclosed Vehicle Spaces on Ro-ro Passenger Ships

**Junzhong Bao \*, Zhijie Bian, Bitong Li, Yan Li and Yuguang Gong**

Marine College, Dalian Maritime University (DMU), Dalian 116026, China; bzj18242092253@163.com (Z.B.); lbitong@163.com (B.L.); lilyli@dlmu.edu.cn (Y.L.); gongyuguang@dlmu.edu.cn (Y.G.)
\* Correspondence: baojunzhong@dlmu.edu.cn

**Abstract:** This study probes the probabilistic features of major fire hazards in enclosed spaces to establish their importance to the occurrence of fires onboard Ro-ro passenger ships and, in turn, to raise effective operational countermeasures. Distinct from the previous studies, the present research employs Bayesian Network (BN) analysis to determine the probabilities of fire hazards more effectively. The results of the first 10 important basic events obtained by the BN model are divided into five groups (Vehicle electrical fires, Reefer vehicle fires, Vehicle-carried cargo fires, Potential causal factors of fire for LIB vehicles, and Vehicle fires originating from human factors), Which prompts the authors to propose preventive measures for mitigating the possibility of fire occurrence on this type of electric vehicle. It is hoped that these measures can be essential justifications for establishing relevant rules regarding carrying LIB vehicles in enclosed spaces on an international level.

**Keywords:** risk analysis; fire hazards; Bayesian Network; Ro-ro spaces

## 1. Introduction

Ro-ro passenger ships (hereafter Ro-pax) on short-sea routes have been extensively utilized in marine transportation, with significant commercial success achieved [1]. However, disturbing accidents, in particular fires and explosions ignited in the cargo spaces of Ro-pax, have made safety issues a prominent concern.

The unique features of Ro-pax may partly contribute to their high fire risks. For example, their large and open garage space(s) for vehicles to roll on or off the ferry can make confinement of fire to its original place difficult [2]. Consequently, the fire starting from an enclosed cargo deck may extend to other areas affecting passengers and crew, leading to more severe consequences [3]. And reportedly, the fire incidents on Ro-pax are not diminishing [4]. In addition, ferries have particular risks from cargo containing combustible materials, such as cars, trucks, and refrigerated containers.

The fire accident on Ro-pax Zhonghua Fuqiang on 19 April 2021, is one of the typical casualties, sounding the alarm that the root causes of fire may not have been identified in previous studies [5]. Similarly, though with no crew injuries, the fire aboard the Ro-ro cargo vessel Höegh Xiamen on 4 June 2020, caused nine firefighters to be injured [6]. These grave accidents highlight that fire casualties in vehicle spaces of Ro-pax entail further in-depth examination.

The analysis studies on more recent fire accidents [7–9] show that the number of fires on Ro-ro decks remains high. Considering that the Fault tree analysis can only accommodate input of objective probability, while the BN method is employed to integrate objective possibility with subjective assessment (condition possibility table), a subjective assessment of occurrence is needed in this study; therefore, BN is preferred.

In this study, firstly, typical causality chains and common fire causes are identified. This is achieved by the literature review, which reexamines selected investigation reports of fire accidents in enclosed spaces onboard Ro-pax. Secondly, the techniques of BN are

applied to underline the priority of causality chains. Thirdly, practically feasible solutions are proposed to reduce the high-profile hazards of fires in enclosed spaces.

Our study focuses on fire hazards that occur in the enclosed vehicle spaces of Ro-pax, and fires occurring in other spaces aboard those ships are excluded. All the fire accidents surveyed in this study are categorized as serious accidents as per established criteria [10,11].

Quantitative risk analysis for the fire hazards in enclosed spaces is performed, and weighted causal chains of fires are prioritized. The outcome of this study can be referenced by competent authorities in developing administrative regulations for safe vehicle transport at sea. In addition, owners and operators of the Ro-ro shipping industry can utilize the results when formulating a plan for identifying key failures and fire causes on board Ro-pax as well as developing measures to improve fire safety in routine operations.

The remainder of this paper proceeds as follows: Section 2 presents a literature review of current studies and their limitations. Sections 3 and 4 lay out the methodology and calculation process applied in this study. Section 5 discusses and analyzes the prioritized factors for fire causes and their corresponding effective risk control options (RCOs). Section 6 presents the conclusion and discussion.

## 2. Literature Review

### 2.1. Current Studies on Fires in Vehicle Spaces on Ro-pax

Several prominent previous risk analysis studies focusing on generic risk analysis of Ro-pax fires were primarily performed in accordance with the FSA Guidelines issued by IMO [12]. The pioneering studies were conducted by DNV and Denmark using Event Tree (ET) models [3,13,14]. Two successive studies (FIRESAFE I in 2016 and FIRESAFE II in 2017) [8] were commissioned by EMSA, with the former focusing on electrical fire as an ignition risk and fire extinguishing failure in Ro-ro spaces, while the latter covered subject matters beyond the coverage of the former. However, to the best knowledge of the authors of this paper, it is still a qualitative and semi-quantitative risk analysis.

Following a study finalized in 2005, DNV GL carried out a follow-up study to examine subsequent fire accidents between 2005 and 2016 to improve fire safety in daily operations from the perspectives of owners and operators through an investigation of common causes of fires in Ro-ro spaces. And those common causes or ignition sources of fires identified have been referenced in this study [15,16].

In the more recent study, the LASHFIRE project, the Failure Mode and Effects Analysis (FMEA) was used to identify sources of fire initiation and hazards worsening the consequences of fires in Ro-ro spaces, and additionally, a list of fire causes, fire origins, failure modes, and safety measures was created (RISE, 2020) [16].

Apart from those above-mentioned studies conducted by organizational entities, professional scholars also probed into the causes of Ro-pax by applying standard risk analysis techniques. Endrina et al. (2018) performed a comparative study of the results of a risk analysis for Ro-pax ships operating in the Strait of Gibraltar with accident statistics covering the period 2000–2011 [17].

Causes or factors leading to fire and explosions onboard ships have been investigated by a few studies from different perspectives and levels. With respect to causal factors involving ship equipment damages, Kwiecinska (2015) [18] identified some detailed causes, including damage to electrical equipment and cables, damage to mechanical equipment, damage to the ship's hull or its equipment, damage caused by external factors, damage occurring during maintenance work/repairs, and spontaneous ignition of cargo. In terms of fires caused by dangerous cargo Ugurlu (2016) [19] presented several fire hazards in tanker transportation of hazardous liquid cargoes, such as hot work, electric arcs, static electricity, and combustible gas accumulation. With regard to categorization of causal factors, Baalisampang et al. (2018) [20] broadly presented four main causal categories, namely human error, mechanical failure, thermal reaction, and electrical faults, which respectively represent 43%, 22%, 14%, and 9% of the fire accidents reviewed. They also identified hot metal surfaces, static electricity, and electrical sparks as the major sources of ignition. And

to classify causal factors in a more specific manner, Wang et al. (2021) [21] studied the critical factors of ship fire accidents at a fine-grained level, for example, refining "human error" to "watchkeeper error, lack of expertise, and poor communication", and "mechanical failure" to "pipeline break, valve leakage, oil tank leakage, and fire extinguishing system failure". All these causal factors have offered insights into the present research on the cause of fire accidents in vehicle spaces on Ro-ro passenger ships.

The main causes of fire identified in previous studies include electrical faults, mechanical failures, thermal reactions, and human error [20,22]. In this study, fire hazards in Ro-ro spaces are classified into four streams: technical failures, which target ship equipment failures and electronic failures leading to vehicle fires; ship cargo hazards, which involve vehicles carried onboard and cargo units loaded on vehicles; vehicles' latching failures; and human factors observed, such as vehicle drivers' unsafe behaviors. These streams constitute the four branches of the fault tree (FT) framework.

The fire casualty data surveyed in the previous studies addressing fire on Ro-pax are mainly derived from the Lloyds Maritime Information Unit database (IHS), the British MAIB database, EMCIP, and GISIS MCI data (FSI 21/5). All these databases constitute the principal sources for statistical collection in the present study. Another source of data for this study is published literature written by Chinese scholars or maritime investigation reports issued by the Chinese government to record some Ro-pax fire accidents that happened in China in the last 20 years, from 2002 to 2021.

In estimating the occurrence of fire accidents in Ro-pax, RISE (2020) [16] reviewed the previous studies in which outputs came up with the frequencies of fire accidents in Ro-ro spaces. Hence, a summary list is produced with the frequencies of fire accidents for various studies [23]. Those frequencies of ship years served as an input in identifying hazards in the LASHFIRE project. In addition, those frequencies can be utilized as a cross-reference for the occurrence of top events in the event tree for future studies to verify future analyses for fire accidents in Ro-pax studies. The input values to those models applied by previous studies were based on statistics from historical data, research findings, and expert judgment.

### 2.2. BN in Quantitative Analysis of Ship Accidents

In the fields of safety studies, a fault tree is used to model the relationship between relevant events and can be applied to both qualitative and quantitative analysis [24]. Fault Tree Analysis (FTA) has also won a place in ship accident analysis. Wang et al. (2013) constructed a fault tree to identify the leading basic events and minimal cut sets for fire accidents on a crude oil tanker [25]. Ali et al. (2022) proposed a fault tree model for an empirical study of 62 collision accidents recorded in a high-profile database over 15 years up to 2020 [26]. Ugurlu et al. (2022) used FTA to determine the probability and importance of the primary causes of ship collision accidents [27]. However, there are certain inherent drawbacks relating to the application of conventional FTA; for example, it is impossible to encompass linguistic variables in the failure logic model when handling uncertainties. To compensate for this limitation, fuzzy set theory is introduced into the FTA process.

FTA, as a kind of static analysis instrument, is incapable of updating the status probability [28], but a Bayesian Network (BN) is a factorization of a probability distribution along with a directed acyclic graph [29]. Unlike FTA, BN is created only using expert judgment, factor correlation, or a literature review and cannot determine how failures lead to unwanted events. Therefore, the integration of FTA and BN has become a potential solution for obtaining more accurate estimation of probabilities, and it is expected to minimize method-related constraints, a common problem in applying FTA alone. Several scholars have integrated the FTA into the BN when analyzing ship accidents. Ugurlu et al. (2022) [27] mapped FTA into a BN and used a dynamic risk analysis methodology to analyze the risks of grounding accidents over 15 years, starting in 2005. Wang et al. (2021) [21] used an FTA-BN algorithm to present a framework for identifying critical risk factors for ship fire accidents. Wu et al. (2021) [30] created a data-driven BN model to analyze potential hazards for electric vehicle fire accidents, which took place in China from

2011 to 2018, and highlighted that charging electric cars transported aboard ships would increase the probability of car fire occurrence. Cenk et al. (2021) applied FTA and a Bayesian Network (BN) analysis to establish the risk level by defining the level of relationship among factors, and then evaluated its impact on grounding [31]. Since the occurrence of certain numbers of basic events under FTA may not always be observed or determined, expert judgment may help determine the probabilities of those basic events.

### 2.3. Limitations of Previous Studies

Some previous studies focused on the establishment of high-level risk methods where event tree analysis is initiated from a Top Event (TE), while other studies established fault tree analysis focusing on qualitative analysis of the influence of causal chains of fire onboard ships. However, there are insufficient quantitative studies on fault tree analysis of fire accidents in Ro-ro spaces, much less studies employing BN in risk analysis. This study intends to conduct research with this methodology to delve into the quantitative relationship between accident chains. Furthermore, BN provides advantages for updating the status probability and for predictive and diagnostic calculation capabilities. Hence, the BN approach for quantitative analysis is more accurate both structurally and probabilistically.

## 3. Methodology

In light of the review of previous research, this study is designed to follow the process of FSA stipulated by IMO, including data collection, hazard identification to obtain significant basic events, and BN which is employed to analyze Ro-ro space fire accident causal factors.

### 3.1. Data Collection

The data in this study is derived partly from those provided by the IMO FSA study on fire accidents on Ro-pax that occurred between 2002 and 2012 and partly from globally published relevant data sources concerning fire accidents on Ro-pax from 2012 to 2021 [3,32]. Further, to compensate for the potential insufficiency of both sources, the authors also included in the scope of the study some fire accidents that occurred on Chinese Ro-pax. They are collected from official reports of fire accidents and some widely recognized journal articles recording fire accidents. A total of 62 cases of fire accidents on Ro-pax, Ro-ro cargo ships, and enclosed vehicle spaces are collected and then reviewed from a novel perspective to extract basic events and representative or typical accident causal chains.

Simply put, the fire accidents selected in this study are within the time period of 2002 to 2021 and are from three literature categories, namely, the FSI 21-5 document, international public websites, accident investigation reports published by the competent authority of the government, and authoritative journal articles in China [33]. In this paper, a semi-structured survey of selected Chinese captains and chief officers is conducted to obtain primary data because, with their shipboard work experiences, particularly the experiences of near-miss incidence, these senior seafarers at management level can have a proper subjective assessment of the possibility of accident occurrence.

### 3.2. Fire Hazard Identification

The aim of identifying fire hazards is to discern typical, basic events. Firstly, the previous key studies are reviewed to extract commonly recognized fire hazards, and then the fire accidents collected (as stated in Section 3.1) are reexamined to identify specific fire hazards. Selected fire accidents in Ro-ro spaces onboard ships are reexamined, leading factors for fire causes are highlighted as basis events, and the causal chains of fires are also established. Fire hazards in Ro-ro spaces are categorized into three streams of failures of ship cargo, including human factors, technical failures, and failures of vehicles' lashing. Each stream accommodates a few fire hazards identified, which become the nodes of BN.

### 3.3. Estimation of the Occurrence of Basic Events

To determine the probability of basic events, the shipyears of 8716 from 2002 to 2018 presented in the LASHFIRE report are quoted, and then by averaging the figure, the authors make further estimations of another 3 years (from 2019–2021), thus obtaining a total shipyears of 9741 for the time period of 2002–2021.

The probability for basic events ($P_i$) was calculated using Equation (1) [34,35].

$$P_i = \frac{\sum\limits_{j=1}^{n} f_j(i)}{Y}$$

$$f_j(i) = \begin{cases} 1/X_j & BE\ No.i\ occurred \\ 0 & BE\ No.i\ not\ occurred \end{cases} \tag{1}$$

where $i$ is the number of basic events of FT, $i = 1, 2, 3\dots\dots n$, $X_j$ is the total number of basic events for the $j$ accident, $j = 1, 2, 3\dots\dots n$, $Y$ is the shipyears, $Y = 9741$.

Then, to calculate the probability of the occurrence of intermediate events leading to the TE, Equations (2)–(4) are used [36].

$$P_{or} = 1 - \prod_{i=1}^{n}(1 - P_i) \tag{2}$$

$$P_{and} = \prod_{i=1}^{n} P_i \tag{3}$$

$$P(TE) = 1 - \prod_{j=1}^{k}\left(1 - P\left(MCS_j\right)\right) \tag{4}$$

where $P_i$ denotes the probability of the basic event $i$, $P\left(MCS_j\right)$ presents the probability of the minimum cut set $j$, and $P(TE)$ is the probability of TE.

### 3.4. Construction of the BN

The authors of this paper have reexamined 62 fire accidents in Ro-ro spaces on board Ro-pax between 2002 and 2021. Those fire accidents are documented in a few authentic data-like sources (see Section 3.1). Through reexamining those accidents, typical causality chains are identified, and common causes of vehicle fires are listed to be used as skeletons of the BN. Further, the occurrences of the basic events in those accidents are accumulated to estimate the frequency of occurrences of the basic events per shipyear. The BN is built with four branches, namely vehicle fuel tank leakage, manual failure, technical failure, and cargo failure. Each branch is rooted, with basic events identified. The proposed BN has also been consulted with duly experienced experts from the Chinese domestic ferry shipping sector.

The Ratio of Variation (ROV) and Birnbaum Importance Measure (BIM) are developed in BN to identify critical events, which are calculated by Equations (5) and (6).

$$ROV(X_i) = \frac{P_o(X_i) - P_r(X_i)}{P_r(X_i)} \tag{5}$$

where $X_i$, $P_o(X_i)$, and $P_r(X_i)$ are the number, the posterior probability, and the prior probability of the basic events, respectively.

$$BIM(X_i) = P(T = 1|X_i = 1) - P(T = 1|X_i = 0) \tag{6}$$

where $X_i$ is the number of the basic events, and $T$ is the TE.

### 3.5. Flowchart of the Study

The flowchart of the study is presented in Figure 1.

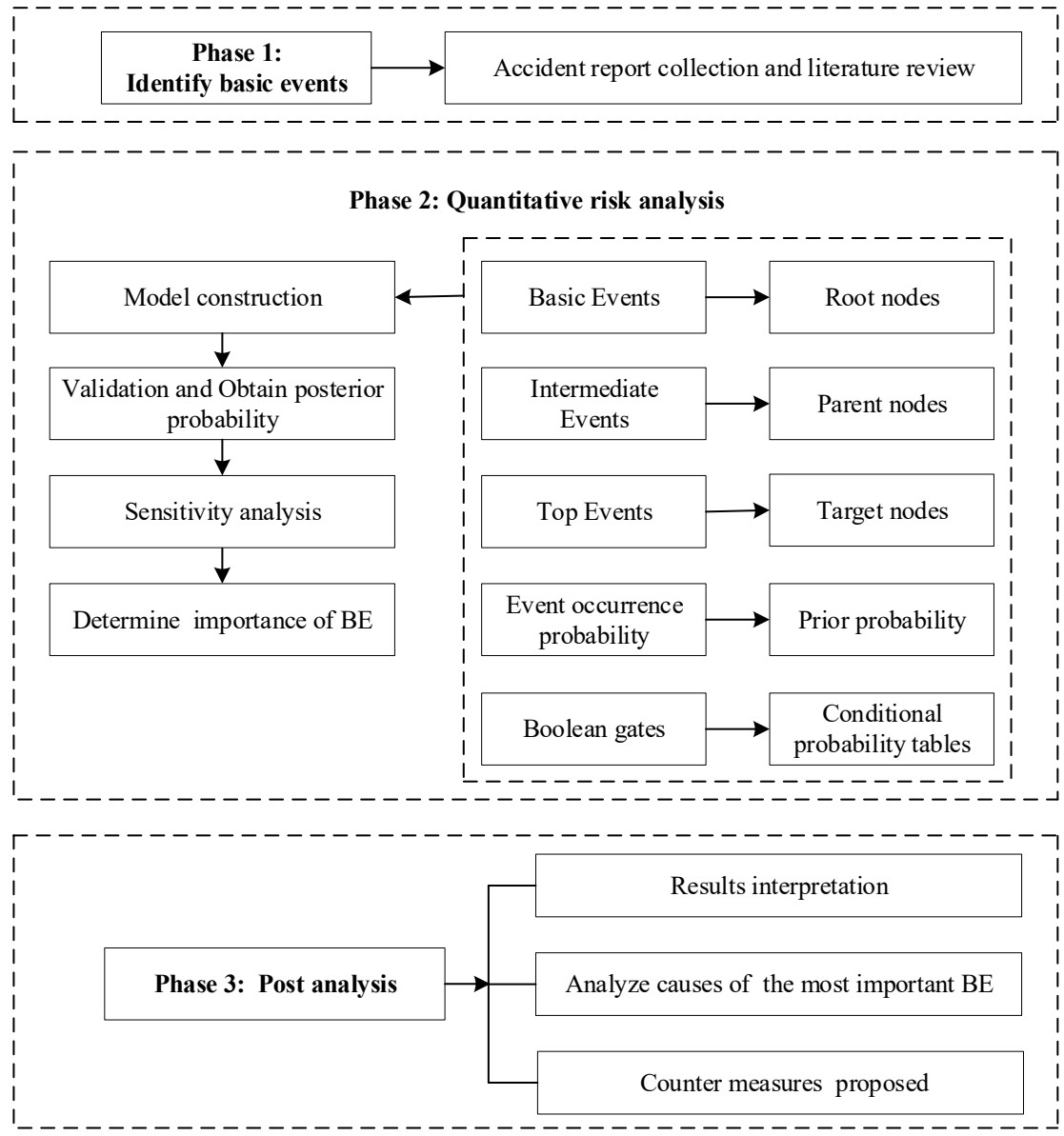

**Figure 1.** Roadmap of the study.

## 4. Case Study

Following the designed methodology, the authors conduct a series of analyses of Ro-pax fire accidents, including BN analysis, BE probability reliability comparison, and the most significant event selection.

### 4.1. BN Analysis of Fires on Ro-pax

In this study, 17 basic events are extracted from the analysis of 62 cases of fire accidents and listed under the three categories listed in Section 3.2 of this paper. The probabilities of Basic Events and the terms of Intermediate Events are shown in Tables 1 and 2, respectively.

A BN model is shown in Figure 2. This model is created using the GENIE software 2.3 to define 24 connections, representing the relationship between 25 nodes, and then a mapping algorithm to quantify the relationship among the variables. In addition, two options of "Yes" or "No" were assigned to each node in the network structure, where a "Yes" status represents the occurrence of the event, whereas a "No" status refers to a non-occurrence condition.

**Table 1.** Probabilities of Basic Events.

| No. | Basic Events | Expected Values (BEs) | Probability (BE) |
|---|---|---|---|
| X1 | insecure lashing/cargo shift | 7 | $7.19 \times 10^{-4}$ |
| X2 | rough seas in heavy weather | 7 | $7.19 \times 10^{-4}$ |
| X3 | electrical boxes short circuit | 1 | $1.03 \times 10^{-4}$ |
| X4 | refrigeration socket transformer malfunction | 1 | $1.03 \times 10^{-4}$ |
| X5 | vehicle engine fire (fuel system fault) | 4.33 | $4.45 \times 10^{-4}$ |
| X6 | vehicle electric fire (electrical equipment defect or short circuit) | 15.83 | $1.63 \times 10^{-3}$ |
| X7 | used car electrical fire | 6.33 | $6.5 \times 10^{-4}$ |
| X8 | reefer unit electrical fire (electrical appliance defects or short circuits) | 8.5 | $8.73 \times 10^{-4}$ |
| X9 | lithium—ion battery—electric vehicles fire | 0.33 | $3.42 \times 10^{-5}$ |
| X10 | discarding non-extinguished cigarette butts | 1 | $1.03 \times 10^{-4}$ |
| X11 | combustible goods left behind | 0.5 | $5.13 \times 10^{-5}$ |
| X12 | staying overnight in cabs | 0.5 | $5.13 \times 10^{-5}$ |
| X13 | operating against the rules or wrongly | 0.83 | $8.55 \times 10^{-5}$ |
| X14 | fire source in vehicle cabs | 0.5 | $5.13 \times 10^{-5}$ |
| X15 | cargo spontaneous combustion | 8.17 | $8.38 \times 10^{-4}$ |
| X16 | cargo burning (nature unknown) | 2.67 | $2.74 \times 10^{-4}$ |
| X17 | dangerous goods burning (undeclared or mis-declared cargo) | 0.83 | $8.55 \times 10^{-5}$ |

**Table 2.** Terms of Intermediate Events.

| No. | Intermediate Events |
|---|---|
| M1 | vehicle fuel tank damage |
| M2 | technical failure |
| M3 | cargo failure |
| M4 | ship power supply equipment |
| M5 | vehicle fires |
| M6 | unsafe behavior of vehicle drivers |
| M7 | cargo fires |

The prior probability of root nodes is the probability of the occurrence of basic events. In addition, Logic gates and expert opinions are used in creating conditional probability tables (CPT(s)), which show conditional probabilities for parent nodes in BN. The frequency of fire accidents before and after the CPT correction is $5.37 \times 10^{-3}$ and $4.31 \times 10^{-3}$, respectively. Tables 3 and 4 show the CPTs of the target node (fire) before and after expert judgment input. Three experts are invited to provide their judgment on corrections to the probability of intermediate nodes. They are all bachelor's degree holders and also hold master's or officer (of management level) competency certificates. They have served on Ro-pax for an average of 20 years. To avoid the possible bias or uncertainties brought about by the subjective judgment of the experts, the triangular fuzzy number is introduced for processing [37,38].

**Table 3.** CPT of the target nodes.

| Node | | | | | | | | | |
|---|---|---|---|---|---|---|---|---|---|
| | M1 | | Yes | | | | No | | |
| | M2 | Yes | | No | | Yes | | No | |
| Fire | M3 | Yes | No | Yes | No | Yes | No | Yes | No |
| | Yes | 1 | 1 | 1 | 1 | 1 | 1 | 1 | 0 |
| | No | 0 | 0 | 0 | 0 | 0 | 0 | 0 | 1 |

**Table 4.** Corrected CPT of the target node (fire).

| Node | | | | | | | | |
|------|------|------|------|------|------|------|------|------|
| M1 | | Yes | | | | No | | |
| M2 | Yes | | No | | Yes | | No | |
| M3 | Yes | No | Yes | No | Yes | No | Yes | No |
| Fire    Yes | 0.98 | 0.95 | 0.95 | 0.95 | 0.95 | 0.9 | 0.95 | 0 |
| No | 0.02 | 0.05 | 0.05 | 0.05 | 0.05 | 0.1 | 0.05 | 1 |

4.1.1. Model Validation

Two fire accidents (See Table 5) are selected as new evidence to certify the applicability of the proposed BN model.

**Table 5.** Synopsis of the accidents.

| Ship Name | Basic Events |
|-----------|--------------|
| Yinghua | (1) Spontaneous combustion of cargo<br>(2) Cargo burning (nature unknown)<br>(3) Dangerous goods burning (Undeclared or mis-declared cargo) |
| Pearl of Scandinavia | (1) Misconduct of vehicle drivers<br>(2) Vehicle electric fire (electrical equipment defect or short circuit) |

By changing the statuses of the basic events involved in two accidents (See Table 5) to "Yes" in the BN model, the probability of fire changed to 0.9311 and 0.8946, respectively. The occurrence of fire in both cases is higher than 0.9, which shows the validity of the BN model used in this paper. The status changes caused by the basic events of the Yinghua fire accident are demonstrated in Figure 2.

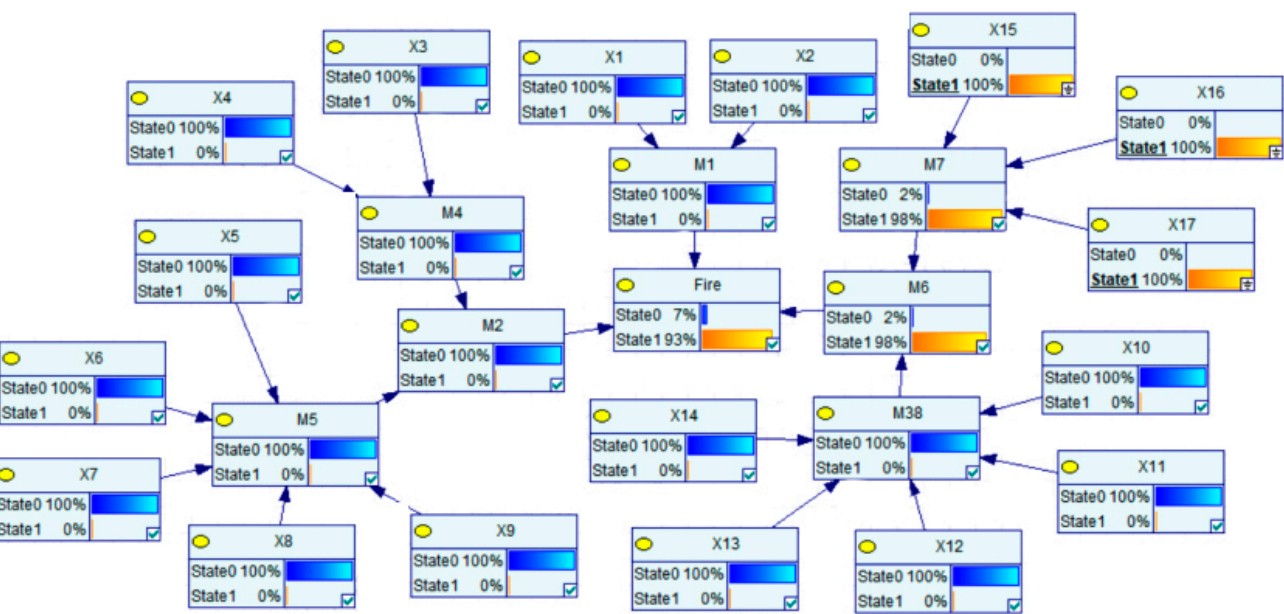

**Figure 2.** Changes in the BN model with new evidence of the Yinghua fire accident.

4.1.2. Sensitivity Analysis

The validated model is then applied to the reasoning process. By setting the target node (fire) as evidence, the posterior probabilities are obtained, as shown in Table 6. Furthermore, the maximum possible causal chain for the accident's occurrence is determined, as shown in Figure 3.

**Table 6.** Results of ranking the importance of the basic events.

| BE | Posterior Probability (BN) | ROV | | BIM | |
|---|---|---|---|---|---|
| | | Result | Rank | Result | Rank |
| X1 | 0.0024 | 2.338 | 16 | 0.010 | 16 |
| X2 | 0.0024 | 2.338 | 16 | 0.010 | 16 |
| X3 | 0.0194 | 187.350 | 6 | 0.806 | 6 |
| X4 | 0.0194 | 187.350 | 6 | 0.806 | 6 |
| X5 | 0.0745 | 166.416 | 10 | 0.717 | 10 |
| X6 | 0.2897 | 176.730 | 8 | 0.762 | 8 |
| X7 | 0.1264 | 193.462 | 4 | 0.833 | 5 |
| X8 | 0.1697 | 193.387 | 5 | 0.834 | 4 |
| X9 | 0.0068 | 199.000 | 2 | 0.860 | 2 |
| X10 | 0.0159 | 153.369 | 14 | 0.663 | 11 |
| X11 | 0.0079 | 153.902 | 12 | 0.662 | 13 |
| X12 | 0.0090 | 175.471 | 9 | 0.757 | 9 |
| X13 | 0.0132 | 154.294 | 11 | 0.662 | 12 |
| X14 | 0.0079 | 153.902 | 12 | 0.662 | 13 |
| X15 | 0.1757 | 208.666 | 1 | 0.899 | 1 |
| X16 | 0.0544 | 197.540 | 3 | 0.851 | 3 |
| X17 | 0.0094 | 109.588 | 15 | 0.473 | 15 |

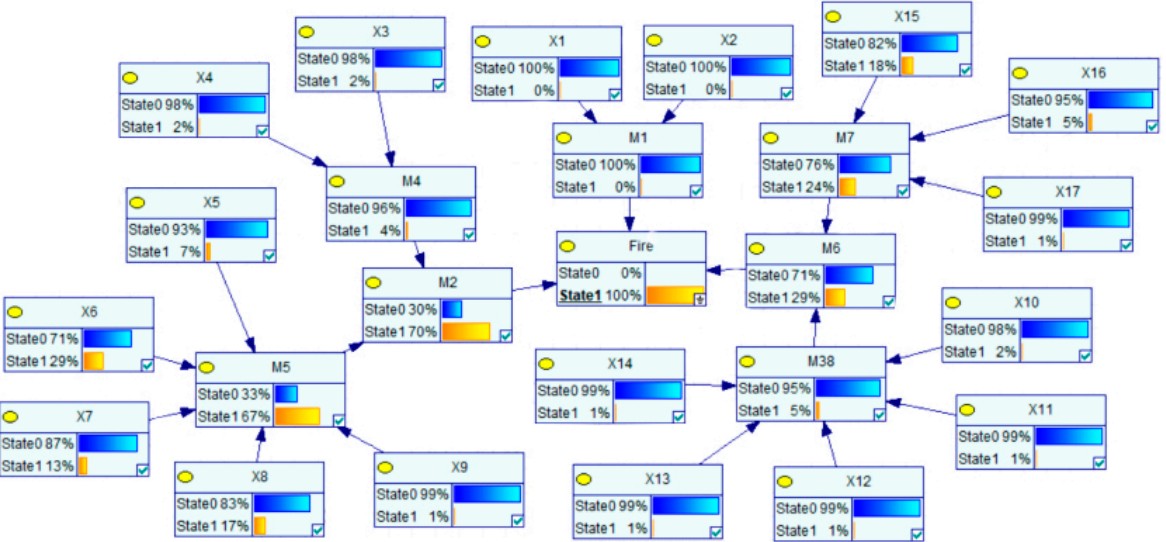

**Figure 3.** The maximum possible causal chain for the occurrence of accidents.

Figure 3 indicates that the route X6—M5—M2—Fire has the highest possibility. Therefore, these events are specially focused. However, the highest possible chain, representing only the formation mechanism of the majority of accidents, cannot exclude other chains. Hence, other highly possible factors such as X5, X7, X8, X15, and X16 are also considered when proposing risk control actions.

In this study, Equations (5) and (6) are used to analyze the most critical basic event. The results are presented in Table 6.

Figure 3 demonstrates the most important basic events leading to TE. It can be observed that such factors as X3, X4, X5, X6, X7, X8, X9, X12, X15, and X16 are the top factors, and further investigation is needed.

### 4.2. BE Probability Reliability Comparison

Findings of frequency as per shipyear for fires from previous studies are summarized in Table 7 (for conventional vehicles) and Table 8 (for HEV, BEV, or Refrigeration unit/RU

vehicles), respectively. Correcting the CPTs brings down the fire probability to $4.31 \times 10^{-3}$, and this value is in the same order but two times higher than that of the DNV-GL study, which is $2.0 \times 10^{-3}$ for 2005–2016. However, it is comparable to that of the FIRESAFE II study, which is $5.28 \times 10^{-3}$ for 2002–2016 (See Table 7). In addition, the probability of BEV is $0.342 \times 10^{-4}$, and this value is in the same order but three times lower than that of the BMVBS study, which is $1.06 \times 10^{-4}$ for 1994–2004. And the probability of RU is $8.73 \times 10^{-4}$, and this value is in the same order but two times higher than that of the BMVBS study, which is $4.94 \times 10^{-4}$ for 1994–2004. (See Table 8).

**Table 7.** Summary of studies addressing Ro-pax fires in Ro-ro space.

| Source | Frequency (per Shipyear) | Time Period (Years) |
|---|---|---|
| [7] | $2.0 \times 10^{-3}$ | 12 |
| [14] | $0.99 \times 10^{-3}$ | 11 |
| [39] | $5.28 \times 10^{-3}$ | 15 |

(Source: RISE, 2020 [16], modified by the authors).

**Table 8.** Fires caused by HEV, BEV, or RU vehicles (not connected/connected to the ship's power distribution).

| Vehicle Type | Scenarios | Number of Fires (per Year) | Frequency (per Shipyear) |
|---|---|---|---|
| HEV/BEV | not connected | 0.3 | $1.06 \times 10^{-4}$ |
| RU | connected | 1.4 | $4.94 \times 10^{-4}$ |

(Source: BMVBS, 2013 [40], modified by the authors).

The differences mentioned above can be attributed to the use of different approaches in the respective analyses. Firstly, while the previous studies all concentrated on the occurrence of top events (fires), which leads to the frequency of fires as per shipyear, this study focuses on the root causes, i.e., the occurrence of basic events is utilized to justify the probability of the top events; simply put, the differences are the result of a different statistical approach. Secondly, more fire accidents are included in the present study, including the 10 fire accidents that happened in Chinese coastal waters from 2002 to 2021, documented in Chinese-language journals, and the recent accidents worldwide from 2017 up to 2021. As claimed by Allianz in 2022, fires have become a consistent loss driver for car carriers over the past decade, and in many cases, fires involving vehicle cargo have resulted in the total loss of cargo and the vessel [41]. Those accident inputs can contribute to the higher probability of fire in this study than in the previous ones.

*4.3. Selection of the Most Important Basic Events*

In this study, the criteria of critical importance, which measure the importance level of respective basic events by their sensitivity and probability, are used to determine the importance level of respective basic events. We adopt the method used in the literature (Vesely et al., 1981) [42] and compute the value of individual basic events importance, as Table 6 (Section 4.1.2) indicates that X15 (Cargo spontaneous combustion) is the most important causal factor of fire accidents in Ro-pax enclosed space, followed by X9 (lithium-ion battery—electric vehicles fire), X16 (cargo burning with unknown reasons), X8 (reefer units electrical fire (electrical appliances defects or short circuit)), and X7 (used car electrical fire).

They are followed by X3 (electrical box short circuit) and X4 (refrigeration socket transformer malfunction) in sixth and seventh place, respectively. And what follows sequentially are X6 (vehicle electric fire (electrical equipment defect or short circuit)), X12 (staying overnight in cabs), and X5 (vehicle engine fire (fuel system fault)). Additionally, the BN model is constructed and verified using GENIE software 2.3. To determine the significance of the order of basic events, both ROV and BIM methods are run, and the

results are approximately consistent. As is shown in Table 6, the first 5 groups of basic events of high possibility are (X15), (X7, X8, X9, X16), (X3, X4), (X6, X12), and (X5).

## 5. Findings and Discussion

According to the results of the first 10 important basic events obtained by the BN model, the most important basic events are X5, X6, and X7 (those related to vehicle fires), X3, X4, and X8 (those related to reefer unit fires), and X15 and X16 (those related to cargo fires). In this section, all these basic events will be discussed in depth.

### 5.1. Vehicle Electrical Fires

This study reveals that vehicle-fire-related basic events X5, X6, and X7 (prior probabilities being $4.45 \times 10^{-4}$, $1.63 \times 10^{-3}$, and $6.5 \times 10^{-4}$ per shipyear, respectively) are among the first ten (posterior probabilities being 0.07, 0.29, and 0.13) of the BN ranking, which indicates that the sources of vehicle electrical fires deserve further investigation.

Electrical faults originating in ships' cargo (vehicles carried onboard) are the most common cause of fires in Ro-ro spaces. According to an IMO study on causes of fire accidents in Ro-ro spaces during the period of 1994 to 2011, electrical fires in vehicles constitute a significant portion. The review of Ro-pax fires in FIRESAFE I shows that approximately 60% of the fires were caused by electrical faults. Vehicles, especially those in poor condition and thereby more prone to electrical faults and leaks, are also a common source of Ro-ro space fires. The symptoms of poor-conditioned vehicles include aging electrical lines, heavy oil stains in the engine compartment, and fuel leaking, which may cause short circuits, sparking, and even engine compartment fire. And short circuits in the vehicle's storage batteries can also cause engine compartment fires.

One effective way to prevent battery short circuits is to disconnect the positive and negative electrodes of the vehicle battery and secure the connecting threads, which can effectively stop the threads from connecting and sparking when the ship vibrates and rolls heavily. Another proper practice is to assign people to inspect the vehicles' electrical systems before they board. In a recent investigation report, the U.S. National Transportation Safety Board (NTSB) recommended that car carriers establish battery securement procedures and a means to ensure that the procedures are followed through adequate oversight of vehicle loading and battery securement. Additionally, the items to be inspected can include identifying any faults in the electrical system that could result in a short circuit or other unintended electrical source of ignition. A Chinese chief mate with 15 years of seagoing service on Ro-pax states in a semi-constructed survey that storage battery short circuits are the major cause of fires, and the five fire accidents he experienced were all caused by them. He further proposes that the most effective action is to disconnect the storage battery power supply and remove the positive and negative electric threads.

### 5.2. Reefer Vehicle Fires

The present study discovers that reefer vehicle-related fires (X8) are the fourth in the BN ranking (posterior possibility being 0.1697), and thus further analysis needs to be made. A study disclosed that the majority of sources of fires started from reefer units, and a significant number of the incidents occurred as a result of electrical fires, particularly relating to refrigerated trailers, though in some cases, fires originated from the ships' own equipment. The root causes are defects in the cables connecting the refrigeration unit with the power supply and sometimes the connection itself. In addition, over one-third of the fires that occurred in Ro-ro space originated in the ship's cargo and were caused by refrigeration units.

While refrigerated units typically constitute merely a rather limited proportion of the cargo carried onboard, they are statistically the most hazardous type of cargo in terms of both hazard probability and severity. Ten participants in the semi-structured survey (four captains and six chief mates) presented their opinions on the causes of reefer vehicle fires, which can be summarized as over-aged engines on reefer vehicles, aging electrical lines,

oil leakage from reefer vehicles, all-time powered cabs, overheated lines due to long-time cooling operations, and automatic initiation of cooling triggered by the temperature rise of the refrigerated cabin during the voyage. This agrees with the RISE statement that electrical faults in refrigeration units are particularly dangerous (RISE, 2020) [16]. Therefore, it can be justified that improving the safety of refrigerated units during transportation will be beneficial to risk reduction in Ro-ro spaces; ship operators need to focus on the more vulnerable and fire-prone refrigeration units connected to the ship's power supply.

The risk control actions proposed by some officers of management level are: inspecting reefer vehicle conditions beforehand to ban those with leaking oil symptoms from boarding the ship; stowing the reefer vehicles properly isolated from other cargo; appointing special attendance to them; conducting regular patrol; standing by fire-fighting appliances during the voyage; and avoiding long-time refrigerating operations to avoid fires caused by power line overheating.

### 5.3. Vehicle-Carried Cargo Fires

This study finds that basic events related to vehicle-carried cargo fires (X15 and X16) are among the first three in the BN ranking (the posterior probabilities being 0.1757 and 0.0544, respectively). This indicates that vehicle-carried cargo fires are worth further investigation.

According to a Chinese captain of a Ro-ro passenger ship in a semi-structured survey, three key causes of ignition for Ro-ro spaces are cargo on vehicles burning, poor vehicle conditions, vehicle cargo shifting caused by rough seas, and improper cargo stowage. Factors relevant to vehicle-carried cargo fires are undeclared or mis-declared cargo and cargo whose nature is unknown to the crew. One captain states that the enormous diversities of the goods make it difficult for the crew to have sufficient knowledge of the nature of the goods.

Special attention should also be paid to the flammable or explosive gases emitted from the burning vehicle-carried cargo (e.g., the chemical reaction of burning silicon mud with sea water can emit hydrogen, which may accumulate in the enclosed space) because they may cause successive explosions. Take two cases of spontaneous combustion of the truck-carried cargo onboard a Chinese Ro-pax, for example. At 2206 LT on 19 April 2021, spontaneous combustion happened on a truck carrying silica mud on board the vessel Zhonghua Fuqiang. At 2231 LT, the master commanded to seal Deck 3 and started the fixed $CO_2$ fire extinguishing system ($CO_2$ released). After the vessel returned to port and berthed, the master evacuated all passengers and most of the crew from the ship. At 0031 LT on 20 April, the shore-based emergency firefighting department took over the firefighting, and at 1141 LT, they initiated opening the sealed space and experienced two consecutive explosions. It is possible that at the initial stage of the firefighting, when the ship was at sea, the space was filled with a large amount of flammable gas and hydrogen produced by the reaction of high-temperature silica mud and hose water, and when the space was reopened, the influx of fresh air mixed with the flammable gases like hydrogen at the stern door led to the first explosion. Furthermore, with the stern door opened wider and the first explosion causing negative pressure inside the space, more fresh air flew into the space. The flammable gases that accumulated between two elevator wells caused another explosion when mixed with the incoming fresh air.

The lesson learned from the above-mentioned cases can be boiled down to the following: firstly, a timely and proper response to the fire can ensure that the ship may return to port to evacuate people from the ship, thus avoiding personal casualties. Secondly, refilling the sealed space with more $CO_2$ from the shore can be an effective way to suffocate the fire and ease the burning [43]. Finally, one of the effective measures to avoid the reoccurrence of similar accidents is that competent authorities inform the front-line operational staff by circulating reports concerning the causes of cargo burning and precautionary measures to take.

*5.4. Potential Causal Factors of Fire for LIB Vehicles*

This study discovers that basic events related to lithium-ion battery (LIB) vehicle fires (X9) rank second in the BN ranking (the posterior probability is 0.0068). A car-maker industry study report reveals that since 2015, the average annual sales growth of global new energy vehicles has been about 54%. Especially in 2021, this increase was recorded at 6.75 million, nearly twice that in 2020. In terms of pure electric vehicles, global sales have reached 4.793 million, doubling the sales in 2020. (Zhan, Y. and Ji, Z., 2022) [44]. In China, this growth was about 157%. In 2021, the volume of sales of new energy vehicles in China accounted for 50% of the global market, with sales reaching 3.52 million, about 2.6 times that in 2020 (Zhan, Y. and Ji, Z., 2022) [44]. It is reasonable to anticipate great growth in demand for transporting new energy vehicles by sea. Meanwhile, it is especially critical for operators to plan their activities carefully concerning vehicle positioning, fire detection, and fighting in storage spaces (McGregor et al., 2021) [2]. Therefore, the root causes of LIB vehicle fires are investigated further.

Generally, the primary cause of BEV/HEV fires is believed to be thermal runaway from LIB. Fires are more likely to occur due to self-ignition (or spontaneous/auto-ignition) in loaded vehicles sustained by abuse such as improper charging. Once the onboard batteries catch fire, it is difficult to suppress it, and in particular, when a LIB catches fire, it is almost inextinguishable, because when the toxic compounds, composed of volatile organic compounds, hydrogen gas, carbon dioxide, carbon monoxide, soot, particulates containing oxides of nickel, aluminum, lithium, copper, cobalt, and hydrogen fluoride, accumulate in the enclosed space, the presence of an igniting source such as a spark or flame, electrical arcs will trigger the explosion, or the compounds may be self-ignited in a poor cooling condition [45].

DNV GL identifies that "shifting cargo represents a risk", which is particularly pertinent to the carriage of electric vehicles. According to the Journal of The Electrochemical Society, one condition leading to LIB thermal runaway is mechanical abuse/lashing failure, which means Electric Vehicle (EV) cargo shifting during the voyage due to lashing failure may lead to a thermal runaway and the ensuing fire [46]. Therefore, giving EV cargo additional lashing to avoid cargo shifting in the sea is a critical action to reduce EV vehicle fires.

Another hazard identified by vessel operators and electric vehicle experts is the risk related to electric vehicles charging without proper authorization [47]. One academic study even highlights that charging EVs onboard may increase the risk of EV fires. Hence, prohibiting charging EV vehicles to avoid thermal runaway and thus bringing down fire risks is also crucial. Additionally, the carriage of damaged electric vehicles can also pose greater fire risks. Therefore, a competent person should thoroughly inspect all electric vehicles before being transported onboard. A suitably qualified person should be assigned to disconnect the battery pack if vehicles are towed or carried by a car transporter.

In a nutshell, the following actions, as proposed by a chief mate with over 20 years of seagoing service experience on Ro-pax, can be taken to reduce the possibility of EV fire: First, EV cargo should be stowed individually under the supervision of personnel during the voyage as per company regulations, with fire-fighting appliances on standby; second, there should be sufficient fire passageway to allow proper ventilation; third, extra lashing should be placed on EV cargo to prevent the vehicle from shifting and colliding when lashings break; finally, bumping and colliding should be avoided when EV cargo embarks or disembarks the ship to avoid physical damage to batteries; and movable fittings in the cargo space should be properly secured to prevent the batteries from being pierced or impacted.

*5.5. Vehicle Fires Originating from Human Factors*

Unsafe behaviors of vehicle drivers (the human factor) are also a causal factor in fire. In the present study, five types of drivers' hazardous behaviors (basic events) are identified, among which the comparatively important ones are discarding non-extinguished cigarette

butts (X10), staying overnight in cabs (X12), and operating against rules or wrongly (X13). In the BN ranking, X12 ranks ninth with a posterior probability of 0.009. One recent fire accident further indicates that drivers staying overnight in cabs can pose a high fire risk [48]. It is worth noting that electric quilts used in cabs in the winter can be a hazard if the power is not completely cut off.

Therefore, the risk control actions for this type of hazard can include the following: observe the company safety supervision regulations strictly to prevent cab drivers and passengers from entering the vehicle space during the voyage; prohibit drivers from staying overnight in cabs; prohibit passengers from carrying flammable or explosive goods; and prohibit people from discarding undistinguished cigarette butts. Fire warning sensors on board are believed to be an effective option to prevent the spread of fire [49,50].

## 6. Conclusions

In this study, 62 fire accidents in enclosed spaces on Ro-pax selected from credential sources are reexamined to identify major fire hazards and establish typical causal chains. The probabilities of basic events are determined as per ship year. Based on these efforts, the top event's probability is figured out, and the critical importance of basic events is prioritized. The comparison of the present study with previous studies indicates that their results are basically at the same level, and this suggests that the results of the present study are acceptable. Those basic events X5 (vehicle engine fire (fuel system fault)), X6 (vehicle electric fire (electrical equipment defect or short circuit)), X7 (used car electrical fire), X8 (reefer unit electrical fire (electrical appliances defects or short circuit)), and X15 (cargo spontaneous combustion) are prioritized by BN and targeted for specific analysis in order to disclose the root causes of such events. A semi-constructed survey involving Chinese senior officers onboard Ro-pax is conducted to sort their opinions on the potential hazards of fires and feasible solutions to reduce the fire hazards on board. In alignment with the findings of the study, some countermeasures are proposed, including disconnecting the storage battery power supply and securing the positive and negative electric threads, avoiding automatic initiation of cooling triggered by temperature rising in the refrigerated cabin during voyages, prohibiting recharging onboard, placing extra lashing on EV cargo, and prohibiting drivers from staying overnight in cabs.

However, in this study, we are unable to construct individual branches of BN for LIB vehicle fires, used car electric fires, and reefer vehicle fires since it has been impossible to determine the probability of occurrence for these three fire events in the case of setting them as immediate nodes. Hence, it is expected to investigate the probability of occurrence for the root nodes of three fire events individually, aiming at constructing a complete BN of fire events onboard a Ro-ro ship by exploring available sources of datasets to determine the probability of occurrence. In addition, to measure the risk level of casualties on Ro-ro passenger ships, PLL (Potential for Loss of Life) is to be calculated; hence, the accumulated probability of fire in enclosed spaces needs to be determined, which is the input (initial frequency) of event tree analysis.

**Author Contributions:** Conceptualization, B.L.; Methodology, Z.B.; Validation, Y.L.; Writing—review & editing, Y.G.; Supervision, J.B. All authors have read and agreed to the published version of the manuscript.

**Funding:** This research received no external funding.

**Institutional Review Board Statement:** Not applicable.

**Informed Consent Statement:** Not applicable.

**Data Availability Statement:** Data will be made available on request.

**Acknowledgments:** The authors would like to thank the editor and the anonymous referees for their valuable comments and suggestions, which have been very helpful in improving the paper.

**Conflicts of Interest:** The authors declare no conflict of interest.

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
