# Peer review of "A Hybrid Approach for Quantitative Analysis of Fire Hazards in Enclosed Vehicle Spaces on Ro-ro Passenger Ships"

_sustainability, doi:10.3390/su151713059_

Round 1
Reviewer 1 Report
This paper examines the probability characteristics of the main fire hazards in enclosed spaces to determine their significance for the occurrence of fires on board Ro-Ro passenger ships. Therefore, it is well suited for the Special Issue "Sustainable Maritime Transportation”. It recommends effective operational countermeasures. Unlike previous studies, this research uses Bayesian Network analysis to more effectively determine the probabilities of fire hazards. The research results include five critical basic events, namely vehicle engine fire, vehicle electrical fire, used car electrical fire, reefer unit electrical fire and cargo spontaneous combustion. In addition, the Bayesian Network analysis also highlights the risk of fires for lithium-ion battery-powered vehicles. The authors try to propose preventive measures to reduce the possibility of fires occurring in this type of electric vehicle, although the recommendations are not fully supported by the Bayesian Network computations. However, it is hoped that the measures proposed can provide essential justifications for establishing relevant rules for the transport of lithium-ion battery vehicles in enclosed spaces at the international level. The paper has formulated a clear and relevant research question. The methodology is appropriate, comprehensible and well described. The paper is well written and the figures and tables are illustrative. The structure is logical and clear. The sources are current but should be cited in the usual manner. The argumentation is convincing and based on the results, and the disadvantages of the proposed method are clearly stated. Overall, the paper is of high quality and has a value for the research field, but there is also room for improvement and further investigation. After correcting the typographical errors, the paper can be published.
The references should be given in the usual form.
Line Hints / Typos
42/43 Every confirmatory statistical method should be investigated, not only BN methods.
94-96 (USCG, 2020;xx McGregor et al., 94 2021; IUMI, 2017; IMO, 2019). Wu et al. (2021) Kwiecinska (2015) (IUMI, 2017) (RISE, 2020). (IMO, 2012)) -> Please, give references in usual form.
135 with a directed a cyclic graph -> with a directed ->acyclic<- graph
192 lasing -> lashing
222 is showed -> is shown
233 Please, examine typesetting.
245 and The terms -> and the terms
257 which shows -> which show
272 the probability of fire increased to 0.9311 and 0.8946 -> 0.8946<0.9
Reviewer 2 Report
Dear authors,
You have made a good effort in conducting a study that would bring benefits to shipping operation and maintain the safety of the marine activity.
However, there are several parts that need serious attentions and improvements. Please refer to my comments.
Thanks.

The paper need to be proofread by qualified proofreader to ensure the clarity of the information in the paper can be delivered accordingly to the readers. It will also help to increase the value and quality of the paper when proper information can be well delivered to the audiences. Please consider to proofread your paper before submitting to the journal. Thanks.
Reviewer 3 Report
Justification for the use of Fault tree analysis with Bayesian Network is presented well, but the claim of BN is suitable for updating status probability is not presented in the manuscript. Authors should demonstrate the status probability updation in their analusis section.
Authors quoted in their literature review that recent study employs FMEA for (sources of fire initiation and hazards worsening 84 consequences of fires in ro-ro spaces, and a list of fire causes, fire origins, failure modes, 85 and safety measures was created.) Why this FMEA is not in the present research? In fact FMEA is an excellent tool with Risk Priority Number as a measure to prioritize potential causes for failures authors should justify why they are not used this tool for identifying the BE or IE.
Reviewer 4 Report
The studies carried out are very relevant and are of interest to readers. But you should check the correctness of the design of references to the literature; Figure 1 - the presentation of BN is a little unfortunate, considering that Figure 2 also shows separate phases; the number of figures should be increased to visualize the materials described in the article; In the conclusions, it is necessary to compare with the results of other scientists.
Reviewer 5 Report
This study has done a lot of work to explore the probability characteristics of major fire hazards in enclosed spaces, and innovatively uses Bayesian network analysis to more effectively determine the probability of fire hazards. In my view, the manuscript could be accepted for the publication in this esteemed journal after incorporation and revision.
The Suggestions are as follows:
(1) In the Introduction, the fourth paragraph mentions “it is paramount to investigate root causes of fires on ro-ro decks using BN methods”. According to the logical relationship, it is reasonable to not find the root cause of the fire and further investigation is needed. However, it is suggested that the author should explain why they chose BN method instead of other methods to investigate.
(2) When abbreviations appear for the first time, the full name should be clearly stated. I noticed that the BN method in the abstract indicates the full name, but the abstract and main text are two parts. The first occurrence of BN method in the introduction should also indicate the full name.
(3) In the sixth paragraph of the introduction, the author mentions this sentence, “Further, the cost and benefit analysis of risk control options proposed are not scoped in this study, neither are emergency response and fire containment issues.” However, there is no need to mention the research that was not done in the article, just focus on your own research content.
(4) The author mentioned “Vehicle fire caused by human factors” in Section 5, and “The findings of the research include five critical basic events (BE) identified namely, vehicle engine fire (fuel system fault), vehicle electric fire (electrical equipment defect or short circuit), used car electrical fire, reefer units electrical fire (electrical appliances defects or short circuit), and cargo spontaneous combustion” in the abstract. But there's a discrepancy between the two parts. Please explain.
(5) In the fifth part, the methods proposed by the author to prevent fires are mostly artificially pre estimated, such as cutting off the power supply, paying attention to refrigeration devices that are prone to fires, and distributing information on the causes of fires to staff. It is recommended to use fire warning sensors to prevent the spread of fires. For details, please refer to “Chemical Engineering Journal 2023, 460: 141661 (https://doi.org/10.1016/j.cej.2023.141661)” and “Chemical Engineering Journal 2022, 431: 134108 (https://doi.org/10.1016/j.cej.2021.134108)”.
It is suggested that the author should further optimize the sentences in the entire manuscript before published.
Round 2
Reviewer 2 Report
Dear authors,
Many parts of my comments have not been responded appropriately. Please take necessary actions to revise and improve the quality of your paper.
Thanks.
Dear authors,
It is recommended that you send your paper to professional proofreader to improve the quality of your writing as well as the presentation of your contents.
thanks.
Reviewer 4 Report
good
Minor editing of English language required
Reviewer 5 Report
The author has carefully made revisions based on the comments of the reviewers, and I believe this manuscript can be published in this respected journal.
Before formal publication, it is recommended that the author continue to optimize their language.
Round 3
Reviewer 2 Report
Dear authors,
The paper should be revised properly as per earliest comments given. As long as the comments were ignored, then the paper is still not ready for publication.
Thanks.
Please proofread before submitted to the journal.